# Predictability and Complexity of Fine and Gross Motor Skills in Fibromyalgia Patients: A Pilot Study

**DOI:** 10.3390/sports12040090

**Published:** 2024-03-25

**Authors:** Nancy Brígida, David Catela, Cristiana Mercê, Marco Branco

**Affiliations:** 1ESDRM Escola Superior de Desporto de Rio Maior, Instituto Politécnico de Santarém, Santarem Polytechnic University, 2040-413 Rio Maior, Portugal; catela@esdrm.ipsantarem.pt (D.C.); cristianamerce@esdrm.ipsantarem.pt (C.M.); marcobranco@esdrm.ipsantarem.pt (M.B.); 2SPRINT Sport Physical Activity and Health Research & Innovation Center, Centro de Investigação e Inovação em Desporto Atividade Física e Saúde, 2001-904 Santarém, Portugal; 3Educação e Treino, Centro de Investigação em Qualidade de Vida (CIEQV), Instituto Politécnico de Santarém, 2001-904 Santarém, Portugal; 4Psicologia Aplicada, Unidade de Investigação do Instituto Politécnico de Santarém, 2001-904 Santarém, Portugal; 5Centro Interdisciplinar de Estudo da Performance Humana (CIPER), Faculdade de Motricidade Humana, Universidade de Lisboa, Cruz Quebrada-Dafundo, 1499-002 Lisboa, Portugal

**Keywords:** fibromyalgia, fine motor control, gross motor control, entropy, single scale, multiscale, IMU

## Abstract

Background: Fine and gross motor tasks are usually used to evaluate behavioral dysfunctions and can be applied to diseases of the central nervous system, such as fibromyalgia (FM). Non-linear measures have allowed for deeper motor control analysis, focusing on the process and on the quality of movement. Therefore, to assess uncertainty, irregularity, and structural richness of a time series, different algorithms of entropy can be computed. The aim of this study was to (i) verify the single-scale and multiscale entropy values in fine and gross motor movements and (ii) to verify whether fine and gross motor tasks are sensitive to characterizing FM patients. Methods: The sample consisted of 20 females (46.2 ± 12.8 years) divided in two groups, an experimental group with 10 FM subjects and a control group with 10 subjects without FM. Inertial sensors were used to collect the finger tapping test (FTT), walking, and sit-and-stand task data. Results: Regarding fine motor skills, patients with FM showed a loss of structural richness (complexity), but they had information processing with greater control in the FTT, probably to simplify task execution and for correction of the movement. On the other hand, people without FM seemed to have more automatic control of the movement when performed with the preferred hand and exhibited similar difficulties to the FM group when performed with the non-preferred hand. Gross motor tasks showed similar entropy values for both groups. Conclusions: The results show that FM patients have movement controls primarily at the level of the motor cortex, whereas people without FM perform movement at the medullary level, especially in fine motor tasks, indicating that the FTT is sensitive to the presence of FM, especially when performed with the preferred hand.

## 1. Introduction

Fine motor control can be defined as the ability to manipulate small objects, manual dexterity, and grapho-motricity. In addition to these capabilities, fine motor control also includes the ability to perform simple, repetitive, and speed-dominated movements, such as tapping a finger on a surface quickly and repetitively [1,2]. A good example of this type of fine motor skill is the finger tapping test (FTT), which is used in healthy people such as in children to assess fine motor skill in fingers [3] and is typically used to assess neurophysiological dysfunctions such as Alzheimer’s disease (AD), Parkinson’s disease (PD), and dementia [4]. According to previous clinical studies, motor and sensory dysfunctions are present in their earliest stages in these diseases [5,6]. This means that there may be a possibility to identify early stages of these conditions with a non-invasive assessment and detect individuals at risk of neurodegenerative diseases [5,6]. Following this, the FTT is a possible and viable test that can be used as a previous indicator for assessing progression and identifying AD, PD, and dementia [6]. Although the FTT is mostly used in neurodegenerative diseases, it has also been applied to diseases of the central nervous system, such as fibromyalgia [7,8].

More recently, fibromyalgia has been assumed to be a disease of the central nervous system that can be characterized by widespread pain, sensitivity to nonpainful stimuli, hypersensitivity to painful stimuli, and fine and gross motor control impairment [8,9].

Recent studies that used the FTT as an assessment tool to evaluate patients with fibromyalgia also used electroencephalography (EEG) and functional near-infrared spectroscopy (fNIRS) to understand what happens in certain areas of the brain and motor cortex when these fine motor movements are performed at a slow or comfortable speed and at maximum speed [7,8]. These studies concluded that there were no differences in motor cortex activation areas in the slow movement speed, but when these FM patients performed the FTT at maximum speed, the test revealed dysfunctional activation and abnormal function of certain motor cortex areas [7,8]. These fine motricity dysfunctions could characterize FM patients, and it is possible that the FTT could facilitate a correct and more detailed diagnosis or even detect specific and different stages of this disease.

Some authors mention that fibromyalgia patients present a functional impairment in gross motor movements, such as gait [9,10,11]. Gait is a highly important task that can provide important information about the patient’s clinical state [12]. Its utility in assessing the ability to receive sensory information, process it, and adapt to new situations is one of the aspects that leads to its widespread use in the motor assessment of different populations. In contrast, the sit-and-stand is a test in which the individual does not need to constantly process sensory information. Therefore, the use of these tests together allows for possible differences in movement control of gross motor skills to be distinguished among fibromyalgia patients.

To be able to characterize different pathological states and identify neurodegenerative or central nervous system diseases, linear analysis may not be enough or adequate. Recently, non-linear measures have allowed for deeper motor control analysis, focusing on the process and on the quality of movement [13]. One of these non-linear analysis measures is entropy [13].

There are many algorithms that can represent entropy on a single scale or on a multiscale [13,14]. Entropy measures on a single scale can be characterized as the loss of information in a time series, and it can be used as a measure of uncertainty and irregularity of time series [13,14]. According to Yentes and Raffalt [14], single-scale entropy is able to quantify the predictability and regularity of the next state of the system. Because of this association between entropy and predictability, when there is higher predictability, the new information you receive from the next states of the system is lower [14]. So, higher values of entropy reveal a higher uncertainty, while less entropy corresponds to lower irregularity or uncertainty in a time series [13]. However, as the name implies, multiscale entropy includes multiple timescales, allowing for a greater understanding of the structural richness of a complex system [14]. A complex movement can be defined as a movement that presents a deterministic origin and a structural richness [14]. With the single-scale entropy measure, patients with specific pathologies or diseases tend to be more predictable, meaning there are lower entropy values [13,14,15]. However, with multiscale entropy, these patients tend to present a complexity loss in some motor tasks, also showing lower entropy values [13,14,15]. To our knowledge, no previous researchers have examined the process of execution; that is, through detailed movement analysis, we intend to bring new insights into possible interferences that the disease has on movement control as a way to monitor the neural processes associated with fibromyalgia. Furthermore, the use of nonlinear techniques applied to time-series data from different sources (e.g., EEG, postural control, etc.) allows for tracking over time (of collection) of different indicators that may enable the assessment of the presence of fibromyalgia [16,17]. Considering this gap in the literature, the following experimental questions were raised: (i) Do fibromyalgia patients exhibit different predictability and complexity in movement control compared to controls during the execution of speed-dominated fine motor tasks? (ii) Are we able to characterize patients with fibromyalgia by using different non-linear algorithms during the finger tapping test? (iii) Is it possible to analyze the predictability and complexity of gross motor movements with daily tasks and characterize fibromyalgia patients?

In order to be able to analyze the execution process, it is necessary to resort to instruments that allow for the collection of more specific data and not just the number of touches on the surface. For this purpose, we used an inertial sensor (IMU) to collect data during the FTT. Inertial sensors allow for the collection of 3D linear acceleration and angular velocity data throughout the entire test in a more practical and precise way [18]. To analyze the entire execution process through linear acceleration, it is necessary to look not only at the acceleration peaks but also at the acceleration time series. But why linear acceleration? In addition to being able to identify and detect acceleration peaks, which translates into touches on a surface, linear acceleration is more sensitive to oscillations, allowing for a more detailed detection of any subtle changes in human movement [18].

According to these statements, the aims of this study were (i) to verify the single-scale and multiscale entropy values in fine and gross motor movements in FM patients and (ii) to verify whether fine and gross motor tasks are sensitive to characterizing FM patients. Therefore, we hypothesize that movement analysis associated with entropy algorithms is appropriate for characterizing the presence of FM.

## 2. Materials and Methods

### 2.1. Sample

The sample was recruited through digital flyers posted on university social media platforms. This method allowed us to reach a diverse and broad audience. Twenty female subjects (Table 1) with ages between 20 and 70 years old were divided into two groups, an experimental group with 10 subjects diagnosed with fibromyalgia by a qualified rheumatologist and according to the standards defined by the American College of Rheumatology (ACR) [19], and a control group with 10 subjects without a diagnosis of fibromyalgia or other diseases, paired by gender, age, preferred hand, height, weight, and physical activity levels with the experimental group. These variables were collected through the application of a survey. All subjects signed an informed consent to participate in the study, which has been approved by the Ethics Committee of the Polytechnic Institute of Santarém (No. 2A-2022 ESDRM).

### 2.2. Procedures

To assess fine motor control, the subjects were asked to perform the finger tapping test (FTT), performing six trials at maximum speed for 10 seconds per trial, starting with the preferred hand and repeating the entire process with the other hand. According to the finger tapping test criteria, the score is the average number of the best five trials [20]. To collect data during the FTT, it was necessary to record all trials with a custom inertial sensor of the MEMS model MPU9250 type, measuring tridimensional (3D) angular velocity and linear acceleration for anteroposterior, mediolateral, and vertical movements (Figure 1).

A rubber finger was used to attach the inertial sensor on the distal phalanx of the second finger. The inertial sensor was used to collect linear acceleration and angular velocity. Data were sent to the computer via Bluetooth and were received at the computer via a serial terminal (connection endpoint) (YAT) [21]. Data were recorded in a text file (.txt). Then, the files were sent to MATLAB version R2021b [22] and SPSS version 29 [23] for data processing.

To assess gross motor control, the participants performed two different tasks, gait for two minutes at a comfortable speed and the 30 s chair sit-and-stand test [24], where they had to stand and sit in a chair as many times as possible for 30 s, with their arms crossed above their chest. These tridimensional linear acceleration and angular velocity data were collected with an IMU (Movesense HR+, Movesense Ltd., Vantaa, Finland), which was placed on the right leg above the malleolus of the fibula for the walking gait task and above the knee for the sit-and-stand task [18] (Figure 2). The IMU data were sent via Wi-Fi to the computer and recorded in an Excel file (.xls). Afterward, the files were sent to MATLAB [22] and SPSS [23] for data processing. To collect fine and gross motor task data, the tests were administered in person always by the same researcher and in the following order to all individuals: (i) FTT, (ii) sit-and-stand, and (iii) gait.

### 2.3. Data Treatment and Statistical Analysis

The .txt files of the FTT and the Excel files of the gait and sit-and-stand tasks were uploaded to MATLAB in a custom script, with the FTT collections trimmed and considered from the first acceleration peak in the finger’s linear acceleration plot, corresponding to the first touch of the finger on the surface, until the 10th second of task execution; the gait and sit-and-stand collections were cut and considered from the first acceleration peak, corresponding to the first step in the gait task and the first stand in the sit-and-stand task, to the last peak of acceleration in each task; linear acceleration was filtered with a digital low-pass filter of the Butterworth type of order 4 and with a cutoff frequency of 30 Hz; the “findpeaks” function in MATLAB was used to detect the touches on the surface, and in this function the definitions of “MinPeakDistance” of 0.120 s and the “MinPeakProminence” of 20% of the acceleration amplitude were applied, which means that the peaks below 120 ms and with less than 20% of the acceleration amplitude were not detected; for FTT, the average time between touches in each trial was calculated; the delay (or tau) and the embedding dimensions were calculated [25]; and the single-scale entropy was calculated by the incremental entropy [14].

In the analysis of very short signals, incremental entropy is a highly effective tool. Being more sensitive, it has the ability to detect subtle changes in the amplitude and structure of signals [26]. The multiscale entropy was calculated by the refined composite multiscale entropy (RCME) with the algorithm by Azami, Rostaghi, Abasolo, and Escudero [13] with the EntropyHub MATLAB toolbox [27]. This algorithm was chosen because it is more recent and can solve different problems in entropy analysis than earlier algorithms. The RCME is faster on long signals and more stable on noisy signals, and can better discriminate elderly from young individuals and patients with neurodegenerative diseases from control subjects [13]. Multiscale entropy analysis was used with 10 temporal scales, and its interpretation was performed through semi-quantitative analysis by observing the plots.

For statistical analysis, the distribution of variables under analysis was tested using the Kolmogorov–Smirnov test and was not assumed for all variables. Therefore, non-parametric tests were performed for comparisons between groups and between trials using the Mann–Whitney U and Kruskal–Wallis tests, respectively. Comparison between hands was performed with the Wilcoxon test. The effect sizes were calculated using Cohen’s d algorithm, according to Fields [28], with 0.2 for a small effect, 0.5 for a medium effect, and 0.8 for a large effect.

## 3. Results

### 3.1. Fine Motor Control

Given that the demand and duration of the FTT protocol could introduce bias in the results of this test (e.g., caused by fatigue), differences between trials of both hands were verified (Table 2).

As a result of this analysis, it was verified that there were no significant differences between trials for the number of touches on the surface, for the mean time between touches, or for tridimensional incremental entropy either for the fibromyalgia group or for the control group. These results show that the demand and duration of the FTT protocol seemed to be adequate for applying the test.

Table 3 presents the values for the statistical analysis and significance level for the variables of number of touches on the surface, time between touches, and mediolateral (ML), anteroposterior (AP), and vertical (V) incremental entropy between hands and per group.

In the fibromyalgia group, there were verified differences in the number of touches, the time between touches, and the anteroposterior incremental entropy. The number of touches was significantly lower in the non-preferred hand and significantly higher in the preferred one (Z = −2.814, *p* = 0.005, r = 0.890). Also, the time between touches and the incremental entropy for the anteroposterior axis was significantly higher in the preferred hand and lower in the non-preferred one (Z = −2.355, *p* = 0.019, r = 0.745 and Z = −2.589, *p* = 0.010, r = 0.819, respectively). In the control group, there were significant differences in the number and time between touches but no differences in incremental entropy. The number of touches was significantly higher in the preferred hand and lower in the non-preferred hand (Z = −3.319, *p* = 0.001, r = 1.000), and the time between touches was significantly higher in the non-preferred hand and lower in the preferred one (Z = −2.746, *p* = 0.006, r = 0.868). The effect size was calculated for the variables with significant differences, showing a large effect [28].

Regarding comparisons between groups, Table 4 presents the values for the statistical analysis and significance level for the same variables: the number of touches on the surface the time between touches, and the mediolateral, anteroposterior, and vertical incremental entropy between groups and per hand.

Although there were no significant differences between the groups or per hand for any of the variables, FM patients showed fewer touches on the surface, more time between touches, and less incremental entropy for mediolateral and vertical movements in both hands compared to the controls. In addition, FM patients showed higher anteroposterior incremental entropy in both hands than the controls.

In addition to the analysis of the previous variables, the refined composite multiscale entropy was also calculated to analyze the complexity level for each hand and axis between the fibromyalgia group and the control group. Regarding the multiscale entropy, the plots below presenting these results represent the entropy levels for each timescale, for each axis of movement, for each hand, and between both groups.

Figure 3a–c refer to the multiscale entropy values for the preferred hand on the mediolateral, anteroposterior, and vertical axes, respectively. These plots also show the comparison of entropy between the fibromyalgia and control groups. In these results, the entropy values are higher in the control group than in the fibromyalgia group for the preferred hand. The scales of entropy values for the preferred hand vary between 0.000 and 0.500, which means that the vertical axis has higher entropy values, followed by the anteroposterior axis and, finally, the mediolateral axis.

Figure 4a–c refer to the multiscale entropy values for the non-preferred hand on the mediolateral, anteroposterior, and vertical axes, respectively. These plots also show the comparison of entropy between the fibromyalgia and control groups.

In these results, the entropy values are higher in the fibromyalgia group than in the control group for the non-preferred hand. Scales of entropy values for the non-preferred hand vary between 0.000 and 0.550, which means that the anteroposterior axis has the lowest entropy values, followed by the mediolateral axis and the vertical axis, with higher entropy values.

### 3.2. Gross Motor Control

The results that are represented in Table 5 show the values for the statistical analysis and significance level for the number of gait cycles and the mediolateral, anteroposterior, and vertical incremental entropy variables between groups and per task.

Although there were no significant differences in incremental entropy values between fibromyalgia and controls in the gait and sit-and-stand tasks, FM patients showed a higher number of gait cycles than the controls.

In addition to the previous single-scale entropy and statistical analysis, the refined composite multiscale entropy was also calculated, which allowed for the analysis of the complexity levels in the gait and sit-and-stand tasks for both groups.

Figure 5a–c refer to the multiscale entropy values for the gait task on the mediolateral, anteroposterior, and vertical axes, respectively. These plots also show the comparison of multiscale entropy between the fibromyalgia and control groups. In these results, the entropy values are similar in both groups, overlapping in several moments of the gait task.

Figure 6a–c refer to the multiscale entropy values for the sit-and-stand task on the mediolateral, anteroposterior, and vertical axes, respectively. These plots also show the comparison of multiscale entropy between both groups.

In these results, like with the gait results, the entropy values are similar in both groups, overlapping in several moments of the task.

## 4. Discussion

This pilot study intended to verify the values of entropy of single-scale and multiscale algorithms during the execution of fine and gross motor control tasks for patients with fibromyalgia and for controls and to compare them between both groups. Additionally, this study aimed to verify whether the finger tapping test, with the use of inertial sensors, allows for the differentiation of characteristics of fibromyalgia patients.

The results point out that the control group performed more touches on the surface than FM patients, and both groups performed a higher number of touches with the preferred hand. In both groups, the time between touches was shorter in the preferred hand (Table 4), and because the controls completed more touches on the surface, the time between touches was shorter in the control group than in the experimental group.

In the incremental entropy or single-scale entropy results, FM patients showed a significantly higher value of entropy in the anteroposterior movements, which indicates that FM patients are less predictable in these movements. This lower predictability means that in the anteroposterior movements, FM patients are more random or less probable with highly new information received from the next states of the system, which is usually seen in younger and healthy adults. With aging and impairment, gait or cyclic movement patterns tend to become more regular or more predictable. In accordance with this tendency, the single-scale results of FM patients also showed lower entropy values in the vertical and mediolateral movements, meaning that they are more predictable when performing the FTT. Nonetheless, when the task demands are increased, like performing it at maximum speed, it might lead to a more irregular pattern, as shown in the anteroposterior movements [14].

Fine motor skills are normally processed in the higher levels of the central nervous system (motor cortex and cerebellum) [29]. In our results, patients with fibromyalgia showed a loss of complexity compared to individuals without fibromyalgia when performing the FTT with the preferred hand. Considering that FM patients not only had lower RCME entropy but also performed fewer touches allows us to state that these patients are spending time receiving and processing feedback information, which allows them to determine movement error and program instructions to reduce that error. In this situation, the system requires more time to process the stimulus and then produce a response, which leads us to suggest that FM patients maintain their information processing conscious and controlled and, therefore, in the presence of a closed-loop model [30]. Continuing this line of thought, Welford [31] suggests that during the processing of the first stimulus, the initiation of the second stimulus must be delayed until the response to the first one begins. This delay is necessary to confirm whether the movement is being executed correctly. Interference will occur if these two signals are processed at the same time. This may be a possible explanation for a longer period of time between touches, as FM patients constantly try to correct the movement but are unable to do so due to the speed of the task, leading to a simplification of the task probably through the freezing of degrees of freedom [32]. In fact, it could explain the loss of complexity verified in our study, confirming a difficulty to adapt similar to what happens in aging and disease states [13]. Another possible reason for the FTT processing of people with fibromyalgia occurring mainly at the level of the motor cortex may be due to central sensitization [33], which is defined as an increased response of the central nervous system to stimuli that are normally not painful. Thus, central sensitization may result in increased activity in the motor cortex during the execution of movements as a way to compensate for the increased sensitivity to pain and discomfort. Although it is already well known that fibromyalgia disease affects the way the brain processes pain and controls movements, our results seem to demonstrate that the motor cortex undergoes demanding processing to compensate for changes in pain perception and brain connectivity.

Alternatively, the control group’s results suggest more automatic behavior during the execution of FTT with the preferred hand. Schmidt, Lee, Winstein, Wulf, and Zelaznik [30] suggests that when the “motor system uses a more automatic control mode and takes advantage of unconscious, fast, and reflexive motor control process, the result is a more effective, efficient, and fluid motion.” Another factor suggesting a more automatic processing is the requirement to perform the task at maximum speed in the same conditions, which seems to not require, for this group, the need for significant corrections to the movement.

Considering the shorter period of time between touches, it seems that the stages of information processing are not involved, meaning that these types of corrections are produced by reflexive mechanisms and that this information is probably processed in the lower levels of the central nervous system (spinal and nerve receptors) [30].

Contrary to the preferred hand, FM patients showed higher complexity in the non-preferred hand than the control group. The possible reason for these results is the fact that the non-preferred hand has less fine motor skills, that is, less dexterity and less coordination ability, than the preferred hand [34]. The controls presented more automatic behavior when executing the FTT with the preferred hand. However, when doing it with the non-preferred hand, they presented greater difficulty in adapting the type of information processing and seemed to pass from processing in the lower levels of the central nervous system (automatic) to controlled processing (higher levels of the central nervous system). This may be a typical characteristic of these populations considering the range of RCME values.

Methodologically, performing the test with both the preferred and non-preferred hands allows us to understand and characterize the fibromyalgia and control groups; hence, it makes perfect sense to apply the test to both hands.

This is one of the advantages of using inertial sensors for the finger tapping test. We cannot collect this amount of information just with the number of touches on the surface, and analyzing the vertical position in the Kinovea software version 0.9.5 does not provide this information, either. It would only be possible if we filmed each attempt in the three planes of motion using two to three cameras, which means much more analysis time. The IMU allows us to carry out a detailed, three-dimensional analysis that is more practical, faster, and cheaper [18].

Otherwise, the gross motor control results show that there were no significant differences between fibromyalgia patients and controls in the single scale. The controls showed fewer gait cycles and a smaller standard deviation, meaning that they performed this task slower and more consistently. FM patients showed more gait cycles, performing this task with more rhythm, but when subtracting the standard deviation from the mean, FM patients actually had fewer gait cycles and were more heterogeneous according to what they felt; that is, they are more different from each other. The controls maintained more or less the same gait pace. Future studies should analyze gait rhythm and stride length. For multiscale entropy, the complexity levels were similar in both groups. This might indicate that entropy in a single scale and a multiscale could be better for analyzing fine motor impairment in these patients and that, when using this non-linear analysis, fibromyalgia could be better characterized by analyzing fine motor control rather than gross motor control.

According to what was referred to above, multiscale entropy seems to be the better option to characterize FM patients through fine motor analysis, and also, because gross motor skills are more automatic than fine motor skills, FTT seems to be a better option.

## 5. Conclusions

In conclusion, although traditional FTT has been successfully applied to people with neurodegenerative diseases, a 3D analysis with an inertial sensor brings new and important information during the execution of the movement and not just the result. These results, combined with non-linear analysis, could allow for a better understanding and characterization of motor control processes for fibromyalgia. With these data, it was possible to verify that FM patients performed a smaller number of touches on the surface and showed a longer period of time between touches in both hands compared to the controls, showing a functional loss in fine motor skills. Fibromyalgia patients also presented lower complexity in the preferred hand and higher complexity in the non-preferred hand compared to the control group. The RCME results suggest that patients with FM demonstrate controlled processing of information during the FTT task execution in both hands in order to simplify the task execution and correct the movement, while the controls have more automatic processing when performing the FTT with the preferred hand and have some difficulties in adapting the type of information processing when performing with the non-preferred hand. The use of inertial sensors to collect data from fine and gross motor tasks has a lot of potential, brings innovation to exercise researchers and professionals, and can also be used in a clinical or practical context. Regarding exercise prescription, it might be a great ally, as it is an easily accessible instrument associated with an app that can calculate the levels of entropy after the test has been applied. These results present the possibility that the FTT with IMU and a non-linear analysis could be used in a clinical context, not only to diagnose or characterize diseases but also to characterize the person and understand whether the exercise prescription needs to be adjusted to improve possible fine motor dysfunctions. The current findings, particularly the observed differences between the FM and control groups, also support the refined prescription guidelines. Given that FM cannot automate and execute highly rapid and repetitive fine motor exercises with fluidity, recommendations for enhancing fine motor skills should not solely rely on exercises that individuals perform at a comfortable pace. Instead, they should incorporate extremely fast and repetitive fine motor exercises, utilizing dedicated apps for this purpose (e.g., CNS Tap, Speed Tapping).

With this information, there is the possibility to prescribe exercise in a more individual and effective way, which should consist of the prescription of movements of larger amplitudes and slower executions, in order to allow for constant control during the execution of the movements, a fact demonstrated by the lack of difference in gross motor tasks.

## Figures and Tables

**Figure 1 sports-12-00090-f001:**
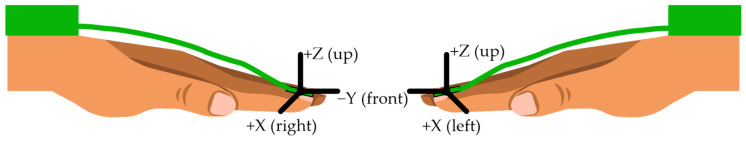
Inertial sensor 3D axis for each hand during FTT execution.

**Figure 2 sports-12-00090-f002:**
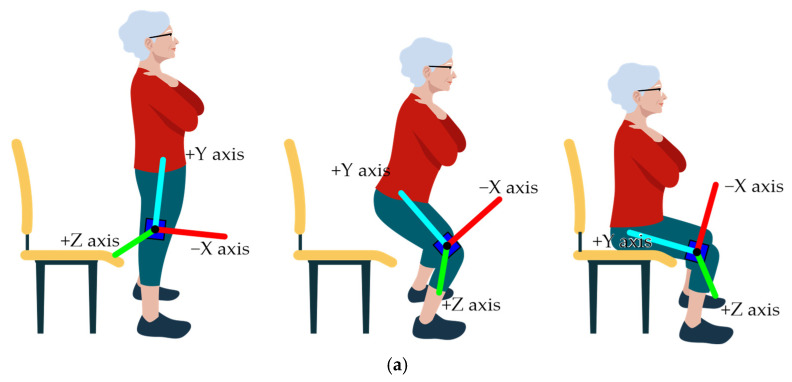
Inertial sensor 3D axis for the (**a**) sit-and-stand and (**b**) walking tasks.

**Figure 3 sports-12-00090-f003:**
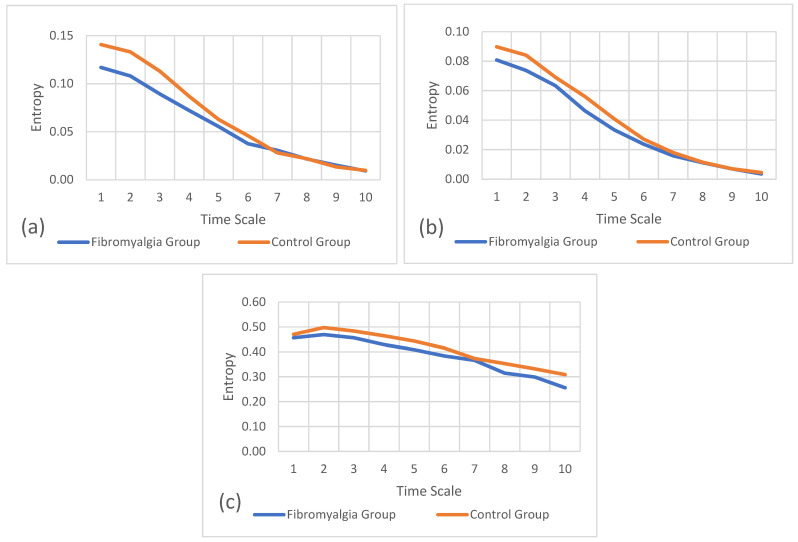
Refined composite multiscale entropy values for the preferred hand between groups. (**a**) Entropy of mediolateral accelerations, (**b**) entropy of anteroposterior accelerations, and (**c**) entropy of vertical accelerations.

**Figure 4 sports-12-00090-f004:**
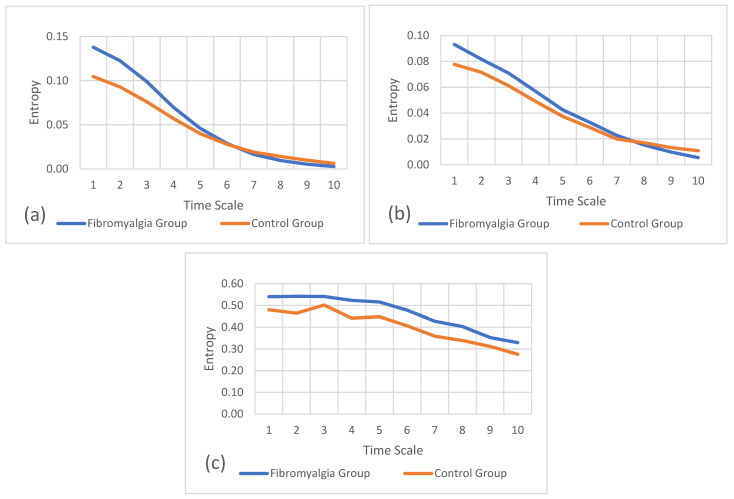
Refined composite multiscale entropy values for the non-preferred hand between groups. (**a**) Entropy of mediolateral accelerations, (**b**) entropy of anteroposterior accelerations, and (**c**) entropy of vertical accelerations.

**Figure 5 sports-12-00090-f005:**
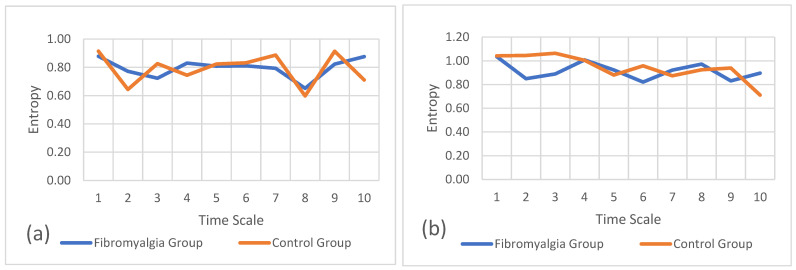
Refined composite multiscale entropy values for the gait task between groups. (**a**) Entropy of mediolateral accelerations, (**b**) entropy of anteroposterior accelerations, and (**c**) entropy of vertical accelerations.

**Figure 6 sports-12-00090-f006:**
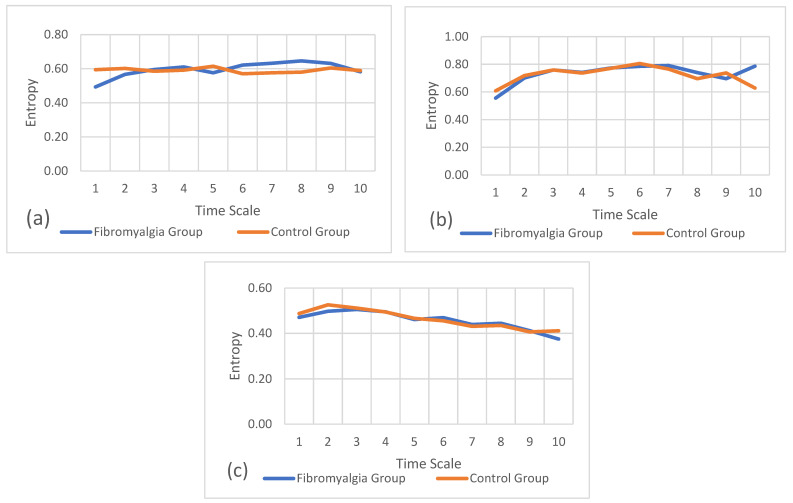
Refined composite multiscale entropy values for the sit-and-stand task between groups. (**a**) Entropy of mediolateral accelerations, (**b**) entropy of anteroposterior accelerations, and (**c**) entropy of vertical accelerations.

**Table 1 sports-12-00090-t001:** Sample characterization.

Group	Age		Height		Weight	
	Mean	SD	Mean	SD	Mean	SD
Fibromyalgia	46.400	12.714	162.900	5.243	63.000	10.536
Control	45.900	12.950	157.800	5.671	60.700	5.675
Total	46.150	12.835	160.350	6.027	61.850	8.540

Note: SD—standard deviation.

**Table 2 sports-12-00090-t002:** Characterization of variables: number of touches on the surface, time between touches, and incremental entropy for the mediolateral, anteroposterior, and vertical axis between trials for each hand and group.

Group/Trial			Trial 1		Trial 2		Trial 3		Trial 4		Trial 5		Trial 6		Test Statistics	
			M	SD	M	SD	M	SD	M	SD	M	SD	M	SD	H	Sig.
Fibromyalgia Group	Preferred Hand	Nbeats	45.0	12.7	46.2	9.6	46.5	10.0	47.0	9.9	46.7	8.0	48.9	9.2	1.15	0.95
		Tbeats	0.2	0.1	0.2	0.1	0.2	0.1	0.2	0.1	0.2	0.1	0.2	0.1	0.93	0.97
		Ent ML	3.9	0.2	4.0	0.2	3.9	0.1	3.9	0.1	3.9	0.1	3.9	0.1	0.68	0.98
		Ent AP	4.0	0.1	4.0	0.1	4.0	0.2	4.0	0.2	3.9	0.2	4.0	0.2	1.94	0.86
		Ent V	3.9	0.2	4.0	0.1	3.9	0.2	4.0	0.2	3.9	0.2	3.9	0.2	1.61	0.90
	Non-Preferred Hand	Nbeats	44.9	9.4	44.4	6.3	44.3	6.3	42.2	9.9	43.4	8.8	46.3	6.3	1.12	0.95
		Tbeats	0.2	0.1	0.2	0.0	0.2	0.0	0.2	0.1	0.2	0.1	0.2	0.0	1.08	0.96
		Ent ML	3.9	0.1	3.9	0.1	3.9	0.1	3.9	0.1	3.9	0.1	3.9	0.1	1.05	0.96
		Ent AP	4.0	0.1	4.0	0.1	3.9	0.1	3.9	0.1	3.9	0.1	3.9	0.1	3.88	0.57
		Ent V	3.9	0.1	4.0	0.1	3.9	0.2	4.0	0.1	3.9	0.1	3.9	0.1	1.13	0.95
Control Group	Preferred Hand	Nbeats	49.6	6.1	47.8	7.8	49.8	5.2	49.5	4.8	47.3	7.5	48.9	3.6	0.56	0.99
		Tbeats	0.2	0.0	0.2	0.0	0.2	0.0	0.2	0.0	0.2	0.0	0.2	0.0	0.89	0.97
		Ent ML	4.0	0.1	3.9	0.1	4.0	0.1	3.9	0.1	4.0	0.1	4.0	0.1	2.99	0.70
		Ent AP	3.9	0.2	3.9	0.2	3.9	0.2	3.9	0.2	3.9	0.2	3.9	0.2	0.78	0.98
		Ent V	4.0	0.1	4.0	0.1	4.0	0.1	3.9	0.1	4.0	0.1	4.0	0.1	1.04	0.96
	Non-Preferred Hand	Nbeats	45.9	5.4	46.0	4.5	46.7	3.4	44.7	6.8	46.5	7.3	48.1	5.0	2.00	0.85
		Tbeats	0.2	0.0	0.2	0.0	0.2	0.0	0.2	0.0	0.2	0.0	0.2	0.0	1.76	0.88
		Ent ML	4.0	0.1	3.9	0.1	3.9	0.1	3.9	0.1	3.9	0.1	3.9	0.1	2.81	0.73
		Ent AP	3.9	0.2	3.9	0.2	3.9	0.1	3.9	0.2	3.9	0.2	3.9	0.2	0.90	0.97
		Ent V	3.9	0.1	4.0	0.1	4.0	0.1	4.0	0.1	4.0	0.1	4.0	0.1	1.20	0.95

Note: Nbeats—Number of touches on the surface; Tbeats—time between touches; Ent ML—incremental entropy for mediolateral axis; Ent AP—incremental entropy for anteroposterior axis; Ent V—incremental entropy for vertical axis; M—mean; SD—standard deviation; H—Kruskal–Wallis; Sig.—significance level.

**Table 3 sports-12-00090-t003:** Characterization of variables for statistical tests and significant values: number of touches, time between touches, and incremental entropy for mediolateral, anteroposterior, and vertical axis between hands and per group. In bold are the variables with significant differences.

Group/Hand		Preferred		Non-Preferred		Test Statistics		
		Mean	SD	Mean	SD	Z	Sig.	Effect Size
Fibromyalgia Group	Nbeats	**46.678**	**9.646**	**44.220**	**7.757**	**−2.814**	**0.005**	**0.890**
	Tbeats	**0.225**	**0.063**	**0.232**	**0.045**	**−2.355**	**0.019**	**0.745**
	Ent ML	3.932	0.142	3.903	0.112	−1.487	0.137	
	Ent AP	**3.964**	**0.144**	**3.925**	**0.101**	**−2.589**	**0.010**	**0.819**
	Ent V	3.940	0.155	3.934	0.127	−0.823	0.411	
Control Group	Nbeats	**48.817**	**5.850**	**46.317**	**5.423**	**−3.319**	**0.001**	**1.000**
	Tbeats	**0.207**	**0.029**	**0.218**	**0.030**	**−2.746**	**0.006**	**0.868**
	Ent ML	3.942	0.098	3.924	0.097	−1.266	0.205	
	Ent AP	3.916	0.164	3.888	0.168	−1.369	0.171	
	Ent V	3.971	0.116	3.975	0.124	−0.861	0.389	

Note: Nbeats—number of touches on the surface; Tbeats—time between touches; Ent ML—incremental entropy for mediolateral axis; Ent AP—incremental entropy for anteroposterior axis; Ent V—incremental entropy for vertical axis; SD—standard deviation; Z—statistical Z test—Wilcoxon; Sig.—significance level.

**Table 4 sports-12-00090-t004:** Characterization of variables for statistical tests and significant values: number of touches, time between touches, and incremental entropy for mediolateral, anteroposterior, and vertical axis between groups and per hand.

Hand/Group		Fibromyalgia		Control		Test Statistics	
		Mean	SD	Mean	SD	U	Sig.
Preferred	Nbeats	46.678	9.646	48.817	5.850	1582.5	0.318
	Tbeats	0.225	0.063	0.207	0.029	1595.0	0.352
	Ent ML	3.932	0.142	3.942	0.098	1757.0	0.945
	Ent AP	3.964	0.144	3.916	0.164	1510.0	0.167
	Ent V	3.940	0.155	3.971	0.116	1617.0	0.416
Non-Preferred	Nbeats	44.220	7.757	46.317	5.423	1477.5	0.120
	Tbeats	0.232	0.045	0.218	0.030	1460.0	0.099
	Ent ML	3.903	0.112	3.924	0.097	1570.0	0.288
	Ent AP	3.925	0.101	3.888	0.168	1668.0	0.588
	Ent V	3.934	0.127	3.975	0.124	1425.0	0.067

Note: Nbeats—number of touches on the surface; Tbeats—time between touches; Ent ML—incremental entropy for mediolateral axis; Ent AP—incremental entropy for anteroposterior axis; Ent V—incremental entropy for vertical axis; SD—standard deviation; U—Mann–Whitney; Sig.—significance level.

**Table 5 sports-12-00090-t005:** Characterization of variables for statistical tests and significant values: number of cycles and incremental entropy for mediolateral, anteroposterior, and vertical axes between groups and per task.

Task/Group		Fibromyalgia		Control		Test Statistics	
		Mean	SD	Mean	SD	U	Sig.
Gait	Ncycles	144.100	29.076	140.000	16.780	42.000	0.579
	Ent ML	3.696	0.187	3.635	0.126	40.000	0.481
	Ent AP	3.977	0.120	4.012	0.132	42.000	0.579
	Ent V	4.110	0.066	4.078	0.067	38.000	0.393
Sit & Stand	Ncycles	32.500	4.403	33.900	5.646	49.000	0.971
	Ent ML	3.609	0.120	3.606	0.134	47.000	0.853
	Ent AP	3.590	0.085	3.509	0.118	35.000	0.280
	Ent V	3.690	0.047	3.713	0.139	37.000	0.353

Note: Ncycles—number of gait cycles; Ent ML—incremental entropy for mediolateral axis; Ent AP—incremental entropy for anteroposterior axis; Ent V—incremental entropy for vertical axis; SD—standard deviation; U—Mann–Whitney; Sig.—significance level.

## Data Availability

Data are available on https://doi.org/10.6084/m9.figshare.25434892 (accessed on 19 March 2024).

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
