# Peer review of "Predictability and Complexity of Fine and Gross Motor Skills in Fibromyalgia Patients: A Pilot Study"

_sports, 2024, doi:10.3390/sports12040090_

Round 1

Reviewer 1 Report

Comments and Suggestions for Authors

ABSTRACT

-The abstract must be completely revised. It is not clear why immediately after the background sentence (i.e., “The finger tapping test (FTT) evaluates … such as Fibromyalgia (FM)”), the authors briefly describe the tests they used and then report the aim of the study. The methods, therefore, appear to be sparse. The results are also unclear. In fact, who would be the patients with fibromyalgia? In the methods it was only reported that 20 females were divided in two groups, an experimental group and a control group.

INTRODUCTION

-The introduction is well-written. The authors cover all the aspects of the research topic comprehensively and in a fluid way for the reader. From fine motor control difficulties to walking impairments in patients fibromyalgia and, moreover, they present the type of data treatment that they carried out (i.e., entropy) by contextualizing it.

-Some statements need bibliographical references.

-“A good example of this type of fine motor skill is the Finger Tapping Test (FTT), FTT evaluates fine motor impairment and is typically used to assess neurophysiological dysfunctions, such as Alzheimer's disease (AD), Parkinson's disease (PD) and dementia”. Fine motor skill is the ability to manipulate objects and is indicative of well-developed neuromotor function and are related to the different activities that people perform throughout their lives. For example, in children the quality of handwriting is highly dependent on manual dexterity. For these reasons, the FFT is also used in other population, in particular in children. I suggest to rewrite this sentence as follows: "A good example of this type of fine motor skill is the Finger Tapping Test (FTT) which, althought it is also used in healthy people such as in children to assess fingers fine motor skill, is typically used to assess neurophysiological dysfunctions, such as Alzheimer's disease (AD), Parkinson's disease (PD) and dementia by evaluating fine motor impairment”. Hence, a part from the reference #3 I suggest adding the following: “Giustino, V., et al. (2023). Manual dexterity in school-age children measured by the Grooved Pegboard test: Evaluation of training effect and performance in dual-task. Heliyon, 9(7), e18327”.

-Please report the hypothesis of the study.

MATERIALS AND METHODS

-Authors should first report the study design. Then they should report the recruitment process (which should not be reported at the end of the paragraph) in detail. They only stated "Subjects were recruited through virtual social platforms and belonged to the same region of Portugal". This is too general and does not allow for replicability. Was the sample drawn from a database? From a register? I don't think anyone randomly contacted people to ask if they had fibromyalgia and if they wanted to participate in the study. After reporting the above, authors can finally report the sample size and how this was divided (the two groups).

-Specify whether a sample size power analysis size was carried out. Based on the design and considering that 2 groups of 10 participants each were included, the minimum power was probably not reached. If so, it should be reported in the title of the manuscript that this is a pilot study.

-Some information regarding the data collection setting should be reported. For example, were the FFT and gross motor control tests randomized across participants or were they administered in a specific order? Similarly, authors should specify this aspect also for the two gross motor control tests administered.

-Moreover, setting details on data collection such as where and who administered the tests should be reported.

DISCUSSION

-Page 11 line 325: "...(Error! Reference source not found.)". Please double check.

-Although practical aspects and strengths and limitations of the research are reported in the conclusion, I suggest rewriting this section leaving the main findings in the “conclusion” paragraph and creating two new paragraphs, that is “Practical impliations” and “Strength and Limitations” of the study.

Author Response

Response to Reviewer suggestions and comments:

ABSTRACT

Reviewer: -The abstract must be completely revised. It is not clear why immediately after the background sentence (i.e., “The finger tapping test (FTT) evaluates … such as Fibromyalgia (FM)”), the authors briefly describe the tests they used and then report the aim of the study. The methods, therefore, appear to be sparse. The results are also unclear. In fact, who would be the patients with fibromyalgia? In the methods it was only reported that 20 females were divided in two groups, an experimental group and a control group.

Author: Thank for your suggestions. The abstract is now revised in the manuscript.

INTRODUCTION

Reviewer: -The introduction is well-written. The authors cover all the aspects of the research topic comprehensively and in a fluid way for the reader. From fine motor control difficulties to walking impairments in patients fibromyalgia and, moreover, they present the type of data treatment that they carried out (i.e., entropy) by contextualizing it.

Reviewer: -Some statements need bibliographical references.

Author: This topic has been revised.

Reviewer: -“A good example of this type of fine motor skill is the Finger Tapping Test (FTT), FTT evaluates fine motor impairment and is typically used to assess neurophysiological dysfunctions, such as Alzheimer's disease (AD), Parkinson's disease (PD) and dementia”. Fine motor skill is the ability to manipulate objects and is indicative of well-developed neuromotor function and are related to the different activities that people perform throughout their lives. For example, in children the quality of handwriting is highly dependent on manual dexterity. For these reasons, the FFT is also used in other population, in particular in children. I suggest to rewrite this sentence as follows: "A good example of this type of fine motor skill is the Finger Tapping Test (FTT) which, althought it is also used in healthy people such as in children to assess fingers fine motor skill, is typically used to assess neurophysiological dysfunctions, such as Alzheimer's disease (AD), Parkinson's disease (PD) and dementia by evaluating fine motor impairment”. Hence, a part from the reference #3 I suggest adding the following: “Giustino, V., et al. (2023). Manual dexterity in school-age children measured by the Grooved Pegboard test: Evaluation of training effect and performance in dual-task. Heliyon, 9(7), e18327”.

Author: Thank you for your suggestions. This sentence was included in the manuscript, as well as its reference (page 1, line 41-44).

Reviewer: -Please report the hypothesis of the study.

Author: The hypothesis of the study was included in the manuscript (page 3, line 121-123).

MATERIALS AND METHODS

Reviewer: -Authors should first report the study design. Then they should report the recruitment process (which should not be reported at the end of the paragraph) in detail. They only stated "Subjects were recruited through virtual social platforms and belonged to the same region of Portugal". This is too general and does not allow for replicability. Was the sample drawn from a database? From a register? I don't think anyone randomly contacted people to ask if they had fibromyalgia and if they wanted to participate in the study. After reporting the above, authors can finally report the sample size and how this was divided (the two groups).

Author: We appreciate your suggestions. This topic has been revised (page 3, line 127-128).

Reviewer: -Specify whether a sample size power analysis size was carried out. Based on the design and considering that 2 groups of 10 participants each were included, the minimum power was probably not reached. If so, it should be reported in the title of the manuscript that this is a pilot study.

Author: We appreciate your suggestion. The sample size power analysis was not carried out for this study, and so the manuscript is now a pilot study (Title).

Reviewer: -Some information regarding the data collection setting should be reported. For example, were the FFT and gross motor control tests randomized across participants or were they administered in a specific order? Similarly, authors should specify this aspect also for the two gross motor control tests administered.

Author: Thank you for your question. The tests weren’t randomized across participants, they were administered in a specific order that was included in the manuscript (page 4, line 163-165).

-Moreover, setting details on data collection such as where and who administered the tests should be reported.

Author: The test were administered in person always by the first author of the study. This information has been added to the manuscript (page 4, line 164).

DISCUSSION

Reviewer: -Page 11 line 325: "...(Error! Reference source not found.)". Please double check.

Author: Thank you very much for this alert, the error was solved.

Reviewer: -Although practical aspects and strengths and limitations of the research are reported in the conclusion, I suggest rewriting this section leaving the main findings in the “conclusion” paragraph and creating two new paragraphs, that is “Practical impliations” and “Strength and Limitations” of the study.

Author: Thank you very much for your suggestions. In order to respect the submission rules and template provided by the journal, we have to maintain the text as it is.

Please open the attachment to see the revised manuscript (underlined in yellow, are the changes made to the manuscript).

Reviewer 2 Report

Comments and Suggestions for Authors

Overall, this manuscript presents a valuable and insightful investigation into the motor control processes in fibromyalgia patients, utilizing innovative methodologies such as 3D analysis with an inertial sensor and non-linear analysis techniques. The study's focus on both fine and gross motor skills, and its exploration of the Finger Tapping Test (FTT) in this context, adds a novel dimension to the existing body of research on fibromyalgia. The findings regarding differences in motor control between fibromyalgia patients and controls are compelling and have significant potential implications for clinical practice, particularly in exercise prescription and rehabilitation strategies. However, there are areas in the manuscript that would benefit from further clarification, expansion, and contextualization to enhance its overall impact and readability.

Introduction   1. The introduction provides a comprehensive background on fine motor control, the relevance of the Finger Tapping Test (FTT), and its application in fibromyalgia. However, the link between these motor skills and fibromyalgia could be further clarified to strengthen the study's rationale.   2. The paper references pertinent studies but may benefit from a broader review, including more recent research, to establish the novelty of this study.     Discussion     1. The comparison between fibromyalgia patients and controls is well-articulated. However, expanding on why these differences are significant in the context of fibromyalgia pathology would add value.   2. The practical implications of these findings for diagnosing or treating fibromyalgia are touched upon but could be expanded. Discussing how these results could be applied in clinical settings would be valuable. 3. The discussion should integrate more with the broader literature on fibromyalgia, particularly regarding how these findings contribute to or challenge existing knowledge.   Conclusion   1. The potential clinical applications of these findings, particularly in exercise prescription for fibromyalgia patients, are well-noted. However, the conclusion could be enhanced by discussing specific strategies or examples of how these findings can be implemented in clinical practice.   2. Ensure that the conclusions drawn are consistent with the analysis presented in the discussion section. For instance, the conclusion mentions lower complexity in the preferred hand for FM patients, but this point needs to be clearly connected with the earlier discussion. 3.  While the study suggests potential clinical applications, providing more concrete recommendations or guidelines for practitioners would enhance the utility of the findings.

Author Response

Response to Reviewer suggestions and comments:

Reviewer: Overall, this manuscript presents a valuable and insightful investigation into the motor control processes in fibromyalgia patients, utilizing innovative methodologies such as 3D analysis with an inertial sensor and non-linear analysis techniques. The study's focus on both fine and gross motor skills, and its exploration of the Finger Tapping Test (FTT) in this context, adds a novel dimension to the existing body of research on fibromyalgia. The findings regarding differences in motor control between fibromyalgia patients and controls are compelling and have significant potential implications for clinical practice, particularly in exercise prescription and rehabilitation strategies. However, there are areas in the manuscript that would benefit from further clarification, expansion, and contextualization to enhance its overall impact and readability.

Reviewer:

Introduction  

Reviewer: 1. The introduction provides a comprehensive background on fine motor control, the relevance of the Finger Tapping Test (FTT), and its application in fibromyalgia. However, the link between these motor skills and fibromyalgia could be further clarified to strengthen the study's rationale.  

Author: Thank you for your suggestion. The link between the skills and fibromyalgia was reviewed and clarified (page 2; lines 67-73).

Reviewer: 2. The paper references pertinent studies but may benefit from a broader review, including more recent research, to establish the novelty of this study.

Author: We appreciate your suggestion. The novelty of the study was established (page 2; lines 94-101).     

Discussion  

Reviewer: 1. The comparison between fibromyalgia patients and controls is well-articulated. However, expanding on why these differences are significant in the context of fibromyalgia pathology would add value.  

Author: Thank you for your suggestion. We have expanded and adjusted this information (page 13; lines 376-392).

Reviewer: 2. The practical implications of these findings for diagnosing or treating fibromyalgia are touched upon but could be expanded. Discussing how these results could be applied in clinical settings would be valuable.

Author: We appreciate your suggestion. This information seems more appropriate to be mentioned in the conclusion, and that is what we did.

Reviewer: 3. The discussion should integrate more with the broader literature on fibromyalgia, particularly regarding how these findings contribute to or challenge existing knowledge.

Author: Thank you for you suggestions, we have added and changed a few points in the discussion.

Conclusion  

Reviewer: 1. The potential clinical applications of these findings, particularly in exercise prescription for fibromyalgia patients, are well-noted. However, the conclusion could be enhanced by discussing specific strategies or examples of how these findings can be implemented in clinical practice.  

Author: Thank you for your suggestion. We have rectified this topic (page 14-15; lines 451-453).

Reviewer: 2. Ensure that the conclusions drawn are consistent with the analysis presented in the discussion section. For instance, the conclusion mentions lower complexity in the preferred hand for FM patients, but this point needs to be clearly connected with the earlier discussion.

Author: Thank you for the alert. It was a writing mistake, which has already been corrected (page 14; line 442).

Reviewer: 3.  While the study suggests potential clinical applications, providing more concrete recommendations or guidelines for practitioners would enhance the utility of the findings.

Author: We appreciate your suggestion. We added some recommendations (page 15; lines 455-461).

Please open the attachment to see the revised manuscript (underlined in yellow, are the changes made to the manuscript).

Round 2

Reviewer 2 Report

Comments and Suggestions for Authors

I have now had the opportunity to review the revised version of your manuscript. I am pleased to inform you that the changes you have implemented have successfully addressed my previous concerns. Your efforts in revising the manuscript are evident and highly appreciated.

The additional information, clarification of key points, and adjustments made in response to the feedback have significantly enhanced the quality and readability of your paper. These revisions have not only strengthened your arguments but also increased the potential impact of your work on the field.

As such, I believe that your manuscript is now ready for publication and I am recommending its acceptance in its current form. It is a valuable contribution to the area of study and I am confident that it will be well-received by the academic community.

Thank you for considering my suggestions and for your dedication to enhancing the quality of your work. I wish you success in your future research endeavors.

Author Response

Dear Reviewer,

We would like to express our sincere gratitude for your time, effort, and expertise dedicated to reviewing our manuscript. Your insightful comments and constructive suggestions have been invaluable in enhancing the quality and clarity of our work. We deeply appreciate your comments and your recommendation for acceptance. 

Thank you.

Best Regards.